# Time-Lag Response of Landslide to Reservoir Water Level Fluctuations during the Storage Period: A Case Study of Baihetan Reservoir

Zhengrong Yang [1], Wenfei Xi [1,2,*], Zhiquan Yang [2,3,4,*], Zhengtao Shi [1], Guangcai Huang [5], Junqi Guo [1] and Dongqing Yang [6]

1 Faculty of Geography, Yunnan Normal University, Kunming 650500, China; zryang@ynnu.edu.cn (Z.Y.); shizhengtao@ynnu.edu.cn (Z.S.); junqiguo@ynnu.edu.cn (J.G.)
2 Key Laboratory of Early Rapid Identification, Prevention and Control of Geological Diseases in Traffic Corridor of High Intensity Earthquake Mountainous Area of Yunnan Province, Kunming 650093, China
3 Faculty of Public Safety and Emergency Management, Kunming University of Science and Technology, Kunming 650093, China
4 Key Laboratory of Geological Disaster Risk Prevention and Control and Emergency Disaster Reduction of Ministry of Emergency Management of the People's Republic of China, Kunming University of Science and Technology, Kunming 650093, China
5 Guizhou Geological Survey Insiture, Guiyang 550081, China; guangcai2020@163.com
6 College of Big Data and Intelligent Engineering, Southwest Forestry University, Kunming 650224, China; yndqyang@outlook.com
* Correspondence: wenfeixi@ynnu.edu.cn (W.X.); yzq1983816@kust.edu.cn (Z.Y.); Tel.: +86-151-9875-6843 (W.X.); +86-150-8708-3552 (Z.Y.)

**Abstract:** Fluctuations in reservoir water levels exert a strong triggering effect on landslides along reservoir banks, constituting a long-term concern in the safe operation of hydroelectric projects and in the prevention and management of geological disasters. While existing research has investigated the impact of periodic water level changes on the deformation of reservoir bank landslides, observation and detection of such deformation are challenging, with noticeable gaps in understanding how these deformations respond to water level changes during the water impoundment period. To address this, our study targets the Baihetan Reservoir, leveraging 567 ascending and descending LiCSAR data and LiCSBAS (the small-baseline subset within LiCSAR) technology to construct a time series of ground deformations in the study area from 2019 to 2023. The TLCC (Time Lag Cross Correlation) model was employed to examine the time-lag response pattern of reservoir bank landslide deformations to reservoir water level changes during the impoundment period. Our findings indicate a clear time-lag response in reservoir bank landslide deformations to water level changes during the impoundment process. The rise in water levels emerged as a primary factor influencing the instability of reservoir bank landslides. During the half-year impoundment period of the Baihetan Reservoir, a time lag of 5–7 days was observed between landslide deformations and increases in water levels, with landslides on the eastern and western banks exhibiting differing time-lag response patterns. Our study illuminates the time-lag effect between water level changes during reservoir impoundment and reservoir bank landslide deformation monitoring. By proposing a quantitative analysis methodology utilizing LiCSBAS technology and the TLCC model, our findings can inform decision-making in the field of disaster prevention and reduction in reservoir engineering.

**Keywords:** reservoir bank landslide deformation monitoring; LiCSBAS technology; TLCC model; lag effect; Baihetan Hydropower Station

## 1. Introduction

Reservoir bank landslides have been confirmed as a common geological disaster phenomenon during the construction and operation of water conservancy and hydropower

facilities [1–3]. A wide variety of reservoir bank landslides, with varying degrees of sliding and diverse manifestations, have been reported in the reservoir areas of hydropower stations under construction or already built worldwide [2,4]. The main factors for the instability of reservoir bank landslides comprise rainfall, reservoir water regulation, or a detrimental combination of the two [5]. Once unstable, they can trigger a series of secondary disasters, damage regional ecosystems, destroy dam bodies and power generation facilities in the reservoir area, and seriously jeopardize the normal operation of hydropower stations and the safety of lives and property of upstream and downstream residents [5–7]. For instance, historical cases such as the landslide at the Malpasset Dam in France in 1959, the Vajont slide in Italy in 1963, and the Zhouqu landslide in Gansu Province, China in 2010, among others, serve as stark reminders of such potential catastrophes [8,9]. Reservoir bank landslides show a strong response to reservoir water storage and regulation, and the deformation of landslides displays a certain delay in response to water level variations, such that human judgments on the movement state of the landslide and predictions on the timing of disaster arising from instability can be affected [10–12]. Accordingly, monitoring the deformation trend of reservoir bank landslides in the reservoir area of hydropower stations, and analyzing the lag effect of the reservoir water level on their deformation, turns out to be a vital aspect of ensuring the safe operation of large-scale water conservancy and hydropower facilities.

Under the effect of the complex terrain and special climatic conditions of the mountainous areas around the reservoir bank of hydropower stations, reservoir bank landslides are generally characterized by high and remote locations, strong concealment, as well as massive potential damage [5,13]. Compared to conventional methods, such as precise leveling measurement, the Global Navigation Satellite System (GNSS), and optical remote sensing technology [1,13], Synthetic Aperture Radar Interferometry (InSAR) technology stands out. The inability of these conventional techniques to facilitate large-scale identification and deformation monitoring is a significant limitation. On the other hand, InSAR technology offers several unique advantages [14]. Primarily, InSAR is capable of wide-range and round-the-clock operations, regardless of the weather conditions. Moreover, it provides stability, high dynamics, precision, and resolution. Owing to these distinct attributes, InSAR has been employed extensively in various fields. Specifically, it has been used in urban ground deformation monitoring [15]. Additionally, it has aided in earthquake analysis [16], volcanic disaster monitoring [17], glacier displacement monitoring [18], and landslide deformation monitoring [11,13]. Each of these applications demonstrates the versatile utility of InSAR technology. InSAR technology employs multi-temporal radar data from repeated orbit observations to detect targets in the interferogram that are capable of providing stable and reliable phase observation values. The time series phase of the interferometric point is analyzed under the condition of phase unwrapping, suggesting the ground time series deformation information; the deformation monitoring accuracy can reach the level of centimeters or even millimeters [13,19,20]. A wide variety of SAR satellites equipped with different bands have been attempted worldwide since the launch of the first L-band SAR satellite, Seasat-A, by NASA in 1978 [21]. With the continuous development of multi-mode, short revisit period, and high-resolution SAR satellites, considerable data have been presented for the theoretical research and application of radar interferometric measurements [14]. Following the free and open radar data from the Earth observation satellite Sentinel-1A/B based on the Copernicus program (Global Monitoring for Environment and Security, GMES) of the European Space Agency over the past few years [22,23], the rich data sources give InSAR technology unique advantages and enormous potential in reservoir bank landslide disaster monitoring. However, large-scale reservoir bank landslide deformation monitoring comprises acquiring, storing, and preprocessing considerable radar data, and inverting a series of time-series parameters, such that a large quantity of processing time is consumed, and high-performance computer and storage space are required [24]. Additionally, due to the special geographical environment and natural climatic conditions where the reservoir bank landslides are located, the application of InSAR tech-

nology in detecting and monitoring reservoir bank landslide deformation will be prone to atmospheric delay errors and phase unwrapping errors [13,24]. Consequently, the accuracy of landslide deformation monitoring results is notably affected, such that a huge challenge is posed to large-scale reservoir bank landslide deformation monitoring. On that basis, Morishta et al. [24]. proposed an open-source InSAR time series analysis method for the automated Sentinel-1 InSAR processor—LiCSBAS (Small Baseline Subset with LiCSAR). LiCSBAS technology is capable of effectively solving the problem of large-scale monitoring requiring considerable processing time while overcoming atmospheric delay errors and control phase unwrapping errors [24,25], thus serving as a novel method for large-scale reservoir bank landslide deformation monitoring.

In general, reservoir bank landslides are affected by reservoir water storage and regulation [26]. Analyzing the lagged response of reservoir bank landslides to reservoir water level variations in conjunction with deformation characteristics takes on critical significance in ensuring the safe operation of water conservancy and hydropower facilities, protecting the life and property safety of upstream and downstream residents in the reservoir area, and contributing to global disaster prevention and reduction efforts. As revealed by numerous case studies on reservoir bank landslide monitoring, most reservoir bank landslides, especially those in large hydropower station reservoir areas, primarily occur in the middle and later stages of reservoir water storage, or even a period of time after the periodic variations in reservoir water levels [26–32]. The above-mentioned phenomena suggest that compared with variations in reservoir water levels, the deformation characteristics or overall instability phenomena of reservoir bank landslides exhibit lag effects. Existing research on the lag effect of reservoir bank landslides has placed a major focus on the following aspects: (1) chart visualization analysis methods [26], which analyze the lag effect by visualizing the deformation time series and reservoir water variations; (2) physical process models [27], which simulate the coupling relationship between reservoir water variations and deformation characteristics from the perspective of the physical mechanism of landslide deformation, using a combination of water-soil factors and geotechnical engineering; (3) function model methods (e.g., the cross-correlation function model, the regression model, and the set pair analysis model [28–30]), calculating the directionality between the deformation time series of reservoir bank landslides and reservoir water level change signals and subsequently determining the degree of lag correlation between the respective phase. However, chart visualization analysis methods can only provide simple qualitative analysis of the lag effect, lacking quantitative data for corroboration. Physical process models have complex parameters and data are difficult to obtain, severely limiting the application of the above-described models. Compared with the above two methods, function models can effectively simulate the degree of correlation between nonlinear time series deformation and reservoir water level variations, analyzing the lag effect of reservoir bank landslide deformation [28]. Nevertheless, function models are dependent on the historical deformation data of reservoir bank landslides. However, due to the challenges in observing and detecting deformation in reservoir bank landslides, existing studies have primarily focused on the effects of periodic water level changes on these deformations. However, research remains significantly lacking concerning the response patterns of reservoir bank landslide deformations to water level fluctuations during the water storage period.

The Baihetan Hydropower Station, located on the second tier of the cascade development (Wudongde–Baihetan–Xiluodu–Xiangjiaba) of the mainstream section of the lower Jinsha River (Figure 1a), is the second largest hydropower project in the world after the Three Gorges Dam. It holds an extremely important position in terms of power generation, flood control, sand blocking, improving downstream navigation conditions, and developing reservoir navigation [33]. Baihetan Hydropower Station began to store water in April 2021, with the reservoir water level rising from 660 m to 825 m, an increase of 165 m. The highest water level during operation is 825 m, and the lowest water level is 765 m, with a water level difference of 60 m. The Baihetan reservoir area spans major active fault zones such as the Zemu River Fault Zone and the Xiaojiang Fault Zone

(Figure 1a). The river valley is deeply cut, the terrain is fragmented, and the construction is intense [13,33]. Meanwhile, water level variations caused by water storage will—to varying degrees—cause erosion, exfoliation, collapse, and landslides, resulting in reservoir bank deformation [10,34]. These conditions pose a serious threat to the safety of the hydropower station's infrastructure and the life and property of residents upstream and downstream. Thus, there is an urgent need to adopt an efficient and feasible method to monitor the movement trend of reservoir bank landslides after the storage of water in the Baihetan reservoir area and effectively analyze its lag effects.

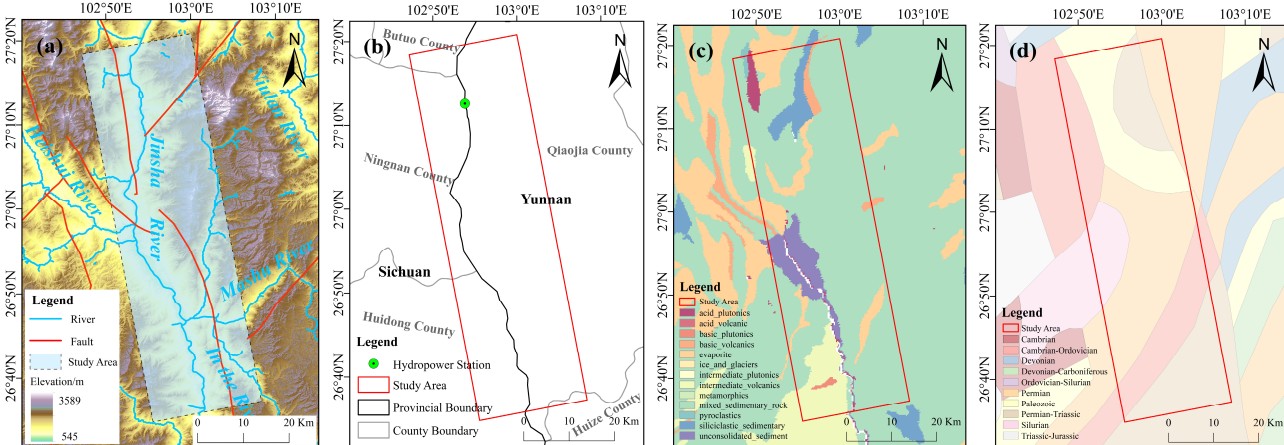

**Figure 1.** Overview of the Research Area. ((**a**) represents the distribution of altitude, faults, and elevation in the research area, (**b**) represents the administrative divisions to which the research area belongs, (**c**) represents the geological lithology of the research area, and (**d**) represents the geological ages of the research area. The elevation data were freely obtained through online access at https://www.eorc.jaxa.jp/ALOS/en/aw3d30/data/index.htm (accessed on 20 March 2023), while the geological data were retrieved through the online shared service platform of the National Earth System Science Data Center of China at http://www.geodata.cn/data/ (accessed on 20 March 2023)).

Given the issues that currently exist in the monitoring of reservoir bank landslide deformation and the analysis of lag effects, there is a lack of in-depth research on the time-lag response pattern of reservoir bank landslide deformation to water level changes during the water storage period. Therefore, this study proposes a method that combines the LiCSBAS (the small-baseline subset within LiCSAR) and TLCC (Time Lag Cross Correlation) technology models to monitor the deformation trend of reservoir bank landslides after the water storage in Baihetan Reservoir and quantitatively analyze the time-lag response pattern of reservoir bank landslides during the water storage period. Initially, the LSM (Layover and Shadow Map) algorithm and the R index are utilized to conduct radar visibility analysis on the ascending and descending track data of the research area, identifying areas of geometric distortion such as shadows, layovers, and foreshortening. Following this, the LiCSBAS technology is employed to procure long time-series deformation information from 2019 to 2022 in the Baihetan Reservoir area. Based on the results of the analysis of the spatial distribution characteristics and temporal evolution laws of surface deformation, the TLCC model is used to quantitatively analyze the lag effect of reservoir bank landslides in the Baihetan Reservoir. Finally, the study delves further into the factors contributing to the lag effect of reservoir bank landslides from a multi-factor perspective. The findings aim to facilitate effective analysis of the time-lag response pattern of reservoir bank landslide deformation during the water storage period and provide scientific evidence for the monitoring and early warning of reservoir bank landslide disasters in large-scale hydropower engineering projects.

## 2. Study Area and Data

### 2.1. Overview of the Study Area

2.1.1. Location of the Study Area

The Baihetan Hydropower Station reservoir area on the mainstream section of the lower Jinsha River (26°34′47.99″~27°20′52.79″ N, 102°47′13.20″~103°08′24.18″ E) was taken as the research area in this study (Figure 1a). The research area was located at the border of Sichuan and Yunnan provinces, spanning Butuo County, Qiaojia County, Ningnan County, Huidong County, Huize County, and Dongchuan District (Figure 1b), and it was distributed in a belt shape along the south–north flow of the Jinsha River. The research area was approximately 207.41 km long and 8.52 km wide, with a total area of 1767.13 km$^2$, located in the northeastern part of the Hengduan Mountains and the southeastern edge of the Qinghai–Tibet Plateau, pertaining to the low-latitude plateau at the Sichuan–Yunnan border [35].

2.1.2. Topography and Terrain

The Baihetan Reservoir area pertains to the high mountain and plateau geomorphologic units of southwestern Sichuan and northeastern Yunnan. The river valley shape of the reservoir bank refers to a wide and gentle "U" shape, with an asymmetric distribution on both sides of the bank slope. The lowest altitude was 545 m, the highest altitude reached 3589 m, and the relative height difference was 3044 m. The slopes on both sides of the reservoir bank were relatively flat at altitudes below 900 m, generally 10°~30°, and steeper above 900 m, roughly 30°~50°. Floodplains, river islands, terraces, and highlands have been extensively distributed in the region, mainly comprising river erosion landforms, tectonic landforms, and glacial erosion landforms, with deep valleys and severe weathering and erosion [13,33].

2.1.3. Hydrological and Meteorological Conditions

The Baihetan Reservoir area belongs to the subtropical plateau monsoon climate, with distinct wet and dry seasons and coinciding rain and heat. Under the effect of the alternating control of the southeastern oceanic monsoon and the polar continental monsoon, the average annual rainfall reached 822.7 mm, the average annual temperature was 21.0 °C, the annual evaporation was 2529.3 mm, the annual radiation was 135.5 kcal/cm$^2$, the annual average sunshine duration reached 2134.2 h [35], the mountain climate features were prominent, and the vertical climate difference turned out to be significant.

The Baihetan Reservoir area is characterized by abundant precipitation, and the water system has been arranged in a feathered pattern, with robust main streams, short tributaries symmetrically distributed on both sides, mainly including the Jinsha River, Yili River, Niulan River, and Heishui River, and other Yangtze River systems [35]. After the reservoir is filled with water, the slopes will undergo long-term soaking of reservoir water, rising and falling water levels, river erosion, and alternating wet and dry cycles, such that a certain degree of slip deformation will be caused, and geological disasters are likely to be triggered.

2.1.4. Stratigraphy and Lithology

The Baihetan reservoir area is located in the low-latitude plateau of Sichuan and Yunnan. The strata have been well developed and cover a wide distribution of weak strata, interlayered soft and hard strata, and loose bodies. In general, the exposed strata comprise the Quaternary series (gravel mixed soil, silt, and clay), Permian (basalt, limestone, and sandstone), Carboniferous (dolomite, limestone, and shale), Devonian (limestone, dolomite, and sandstone), Silurian (mudstone, sandstone, and shale), Ordovician (dolomite and sandstone), Cambrian (limestone, sandstone, and shale), Sinian (dolomite, sandstone, and shale), as well as Pre-Sinian (slate, phyllite, marble, slate, and limestone) strata (Figure 1c,d) [13,33,35].

### 2.1.5. Active Tectonics

The Baihetan reservoir area is located in the fold fault zone of the Yunnan–Guizhou–Sichuan–Hubei depression, the northwestern part of the South China fold system, and the southwestern part of the Yangtze quasi-platform. It is characterized by developed fault structures, strong tectonic movements, and frequent earthquakes. In general, the region is characterized by the presence of active fault zones (e.g., the Zemu River fault zone and the Xiaojiang fault zone), both of which display an approximate north–south trend and left–lateral slip components [33]. The rock and soil bodies in the fault zone are broken, and the joints and fissures are well developed, thus contributing to the occurrence of geological disasters (e.g., landslides, collapses, and debris flows).

### 2.2. Data Set

#### 2.2.1. LiCSAR Data

The main data set employed in this study to monitor the surface deformation trend of the Baihetan reservoir area was the open-source InSAR data—LiCSAR system products released by the Center for Earthquake, Volcano, and Tectonic Observation and Modeling (COMET), with the free access of https://comet.nerc.ac.uk/COMET-LiCS-portal/ (accessed 15 March 2023). The LiCSAR system was based on large-scale interferometric processing of Sentinel-1 data globally, which has been primarily adopted to monitor the temporal evolution and spatial distribution of surface deformation. The LiCSAR system can automatically provide wrapped and unwrapped interferograms, coherence estimation maps, time series, and other products with a resolution of 0.001 degrees (WGS-84 coordinate system) [24,25]. The differential interferometric data provided by the LiCSAR system platform were processed using the InSAR data processing software GAMMA and Snaphu. Data preprocessing and differential interferometry processing were performed using GAMMA software, while phase unwrapping was performed with Snaphu software [24]. The LiCSAR system employs innovative algorithms, processing, and storage solutions to reduce data processing time and required computer disk space. In accordance with priority areas (updated monthly, weekly, or in real-time) [25], LiCSAR products are continuously updated at a certain frequency, providing a new method for large-scale reservoir bank landslide deformation monitoring.

To acquire the long-time series deformation results of the Baihetan reservoir area, 567 LiCSAR data obtained from differential interferometric measurements were selected in this study based on Sentinel-1 data from July 2019 to April 2022. Each data set includes an unwrapped differential interferogram, coherence map, and corresponding DEM image of the area. Among them, 238 ascending orbit data (Frame: 026A_06324_131313) and 329 descending orbit data (Frame: 062D_06231_131313) were used. Table 1 lists the detailed information on the LiCSAR data.

**Table 1.** LiCSAR data parameters.

| Orbit | Frame | Data Time Phase | Number of Images | Imaging Mode | Wave | Wavelength/cm |
|---|---|---|---|---|---|---|
| Ascending | 026A_06324_131313 | 1 July 2019–27 April 2022 | 238 | IW | C | 5.6 |
| Descending | 062D_06231_131313 | 3 July 2019–17 April 2022 | 329 | IW | C | 5.6 |

#### 2.2.2. Auxiliary Data

The auxiliary data comprised the following: (1) Generic Atmospheric Correction Online Service for InSAR (GACOS), GACOS are the atmospheric correction data provided by the team of Professor Zhenhong Li at Newcastle University, used for atmospheric correction of tropospheric noise [36]. These data can be obtained online at http://www.gacos.net/ (accessed 15 March 2023). (2) DEM data; the DEM data uses the digital elevation model with a spatial resolution of 30 m from the Japan Aerospace Exploration Agency's ALOS WORLD 3D. They were adopted to calculate elevation, slope, aspect, curvature, and to remove the effect of terrain phase. They can be obtained online at https://www.eorc.

jaxa.jp/ALOS/en/aw3d30/data/index.htm (accessed 20 March 2023). The elevation, slope, and aspect are calculated using ArcGIS 10.8.1 software with the DEM of the study area as input data. (3) High-resolution Google imagery (Google Earth) is available online at http://www.google.cn/intl/zh-CN/earth/ (accessed 30 March 2023). The data acquired from Google Earth are used to annotate the study area and to overlay the acquired InSAR deformation results for analysis. Table 2 lists the detailed information on the auxiliary data.

**Table 2.** Detailed information on surface deformation impact factors and other data.

| Data Name | Data Time-Phase | Data Type | Data Scale | Data Source |
|---|---|---|---|---|
| GACOS | July 2019–April 2022 | Raster | 90 m | Newcastle University, UK |
| ALOS DEM | 2018 | Raster | 30 m | Japan Aerospace Exploration Agency (JAXA) |
| Elevation | 2018 | Raster | 30 m | Japan Aerospace Exploration Agency (JAXA) |
| Slope | 2018 | Raster | 30 m | Japan Aerospace Exploration Agency (JAXA) |
| Aspect | 2018 | Raster | 30 m | Japan Aerospace Exploration Agency (JAXA) |
| Fractional Vegetation Cover | July 2019–April 2022 | Raster | 30 m | Google Earth Engine |
| Google Images | 2021 | - | 0.2 m | Google Earth |

## 3. Materials and Methods

Figure 2 presents the overall technical process of this study. First, the visible area of the SAR data was analyzed through the local incidence angle, and the shadow and layover areas were masked. Next, using the LiCSBAS technology, the long-term deformation information from July 2019 to April 2022 in the Baihetan reservoir area was obtained, and the spatial distribution characteristics and temporal evolution of the surface deformation in the reservoir area were analyzed. Finally, combining the TLCC model, the lag effect of the reservoir bank landslide in the Baihetan reservoir was quantitatively analyzed, and the factors for the lag effect of the reservoir bank landslide were further explored from a multifactorial perspective.

### 3.1. Analysis of SAR Data Geometric Distortion

Under the particularity of radar side-looking imaging (Figure 3), three types of geometric distortions (i.e., layover, foreshortening, and shadow) were produced in accordance with the incident angle and different topographic features [13,14]. The mountain area of a large hydroelectric dam reservoir was located in a high mountain canyon area. The terrain was complex, and the mountains were steep, such that geometric distortion was generated in some areas of the SAR image. To increase the accuracy of surface deformation analysis results, the correlation between the satellite observation direction and the local incident angle based on the LSM algorithm [37] and the R index [22,38] was developed in this study, as expressed in Equation (1):

$$\begin{cases} \theta < 0°, & Layover \\ 0° \leq \theta \leq 90° \ and \ \theta - \alpha < 0°, & Foreshortening \\ 0° \leq \theta \leq 90° \ and \ \theta - \alpha > 0°, & Visibility \\ \theta > 90°, & Shadow \end{cases} \tag{1}$$

In Equation (1), $\theta$ represents the local incidence angle; $\alpha$ denotes the incidence angle of the Sentinel-1 satellite line of sight (LOS).

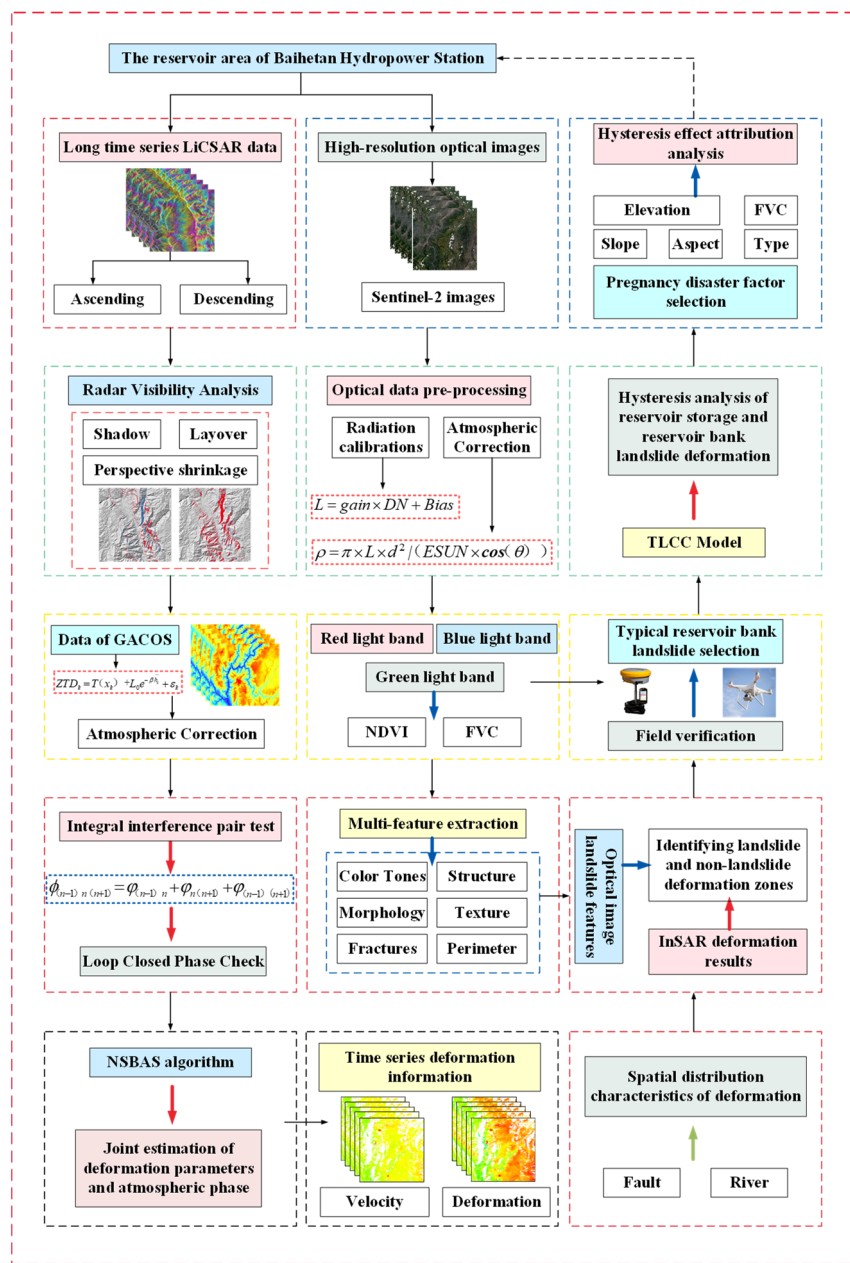

**Figure 2.** Technical roadmap.

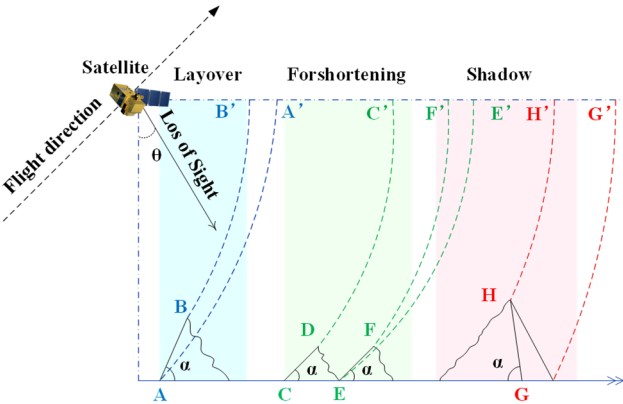

**Figure 3.** Side-looking radar imaging and geometric distortion (*α* represents the local incidence angle, while *θ* denotes the incidence angle of the satellite line of sight).

Figure 4 presents the geometric distortion detection and recognition results of the SAR data covering the research area analyzed based on the local incidence angle. As depicted in the figure, the overlay and shadow areas of the ascending orbit data were concentrated on the east bank of the research area, and the visibility was relatively better than that of the descending orbit data. The geometric distortion areas of the descending orbit data were largely overlays, and they were distributed on both sides of the Baihetan section of the Jinsha River, suggesting that the adjacent mountains were steep in high mountain canyon areas. When geometric distortions (e.g., overlays or shadows) occur, the surrounding area and produced passive geometric distortions will be affected, which cannot be ignored. A large area of perspective shrinkage and overlay will lead to the reduced accuracy of InSAR deformation detection results and hinder the process of interpreting reservoir bank landslides. In addition, it is almost unlikely to detect deformation signals in shadow areas, making the interpretation process of landslides in this area impossible. Accordingly, in the process of reservoir bank landslide identification, ascending and descending data should be combined, and masking should be adopted to eliminate shadow areas to increase the highly applicable areas that are not affected by geometric distortion and increase the accuracy of surface deformation analysis results.

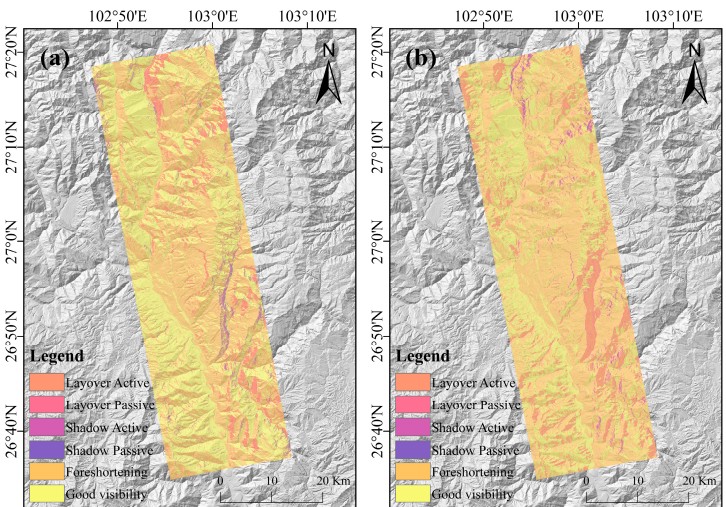

**Figure 4.** Detection and identification of geometric distortion areas in the study area. ((**a**) for ascending orbit results, (**b**) for descending orbit results).

### 3.2. GACOS Atmospheric Correction

GACOS tropospheric delay products were adopted to correct the atmospheric phase arising from the special natural climatic conditions in the Baihetan reservoir area. GACOS employs the European Centre for Medium-Range Weather Forecasts (ECMWF) weather model with a resolution of 0.1° and 6 h to generate a 90 m resolution atmospheric correction map with the Iterative Tropospheric Decomposition (ITD) model [36]. However, not all GACOS products are capable of effectively correcting all interferograms. When correcting some interferometric pairs, it is possible that the atmospheric effect will not be reduced, and the phase error will be introduced. Thus, the phase standard deviation (STD) has generally served as an indicator to evaluate the effect of atmospheric correction [24,25]. In this study, the ITD model was used to separate the layered delay and turbulence delay from the tropospheric zenith total delay (ZTD), where ZTD is defined as follows:

$$ZTD_k = T_{(x_k)} + L_0 e^{-\beta \bar{h}_k} + \varepsilon_k \tag{2}$$

where $ZTD_k$ denotes the zenith total delay at the $k$ position, $T_{(x_k)}$ represents the turbulent component under the $x_k$ coordinates, $x_k$ is the site coordinate component in the local terrain center coordinate system, $L_0$ expresses the sea level layered component delay; $e^{-\beta}$

is the layered component, $\varepsilon_k$ denotes the remaining unmodeled residual error, including unmodeled layered and turbulent signals, and $\bar{h}_k$ is the height scale, where

$$\bar{h}_k = (h_k - h_{min})/(h_{max} - h_{min}) \tag{3}$$

ITD primarily uses Equation (2) to iteratively estimate the height scale function and find the best parameter $(L_0, \beta)$. First, it is assumed that there is no turbulent signal and a pair of $(L_0, \beta)$ is obtained, such that the turbulent signal is identified through the inverse distance weighted regression (IDW) model, and the turbulent signal is removed from the total delay to generate an updated layered delay. The above-mentioned steps are iterated multiple times till a stable coefficient is obtained. Lastly, a set of specified coefficients $(L_0, \beta)$ for the given area is output, along with the turbulence delay and the residuals of the respective GPS station. All converged turbulence delay components and residuals are interpolated to the $k$ position, and the estimated coefficients $(L_0, \beta)$ are introduced into the layered component delay at $L_0 e^{-\beta \bar{h}_k}$, and the two are introduced together to generate the relative ZTD at each position [24], thus completing the removal of redundant atmospheric delay phase. Figure 5 presents the phase standard deviation before and after the introduction of the GACOS tropospheric delay correction product for atmospheric correction. As depicted in Figure 5, the phase standard deviation is significantly reduced after correction with GACOS data compared with prior to correction. As revealed by the mentioned result, in the mountain area of a hydropower station reservoir with significant terrain fluctuations, GACOS data are capable of effectively suppressing the effect of tropospheric atmospheric delay errors.

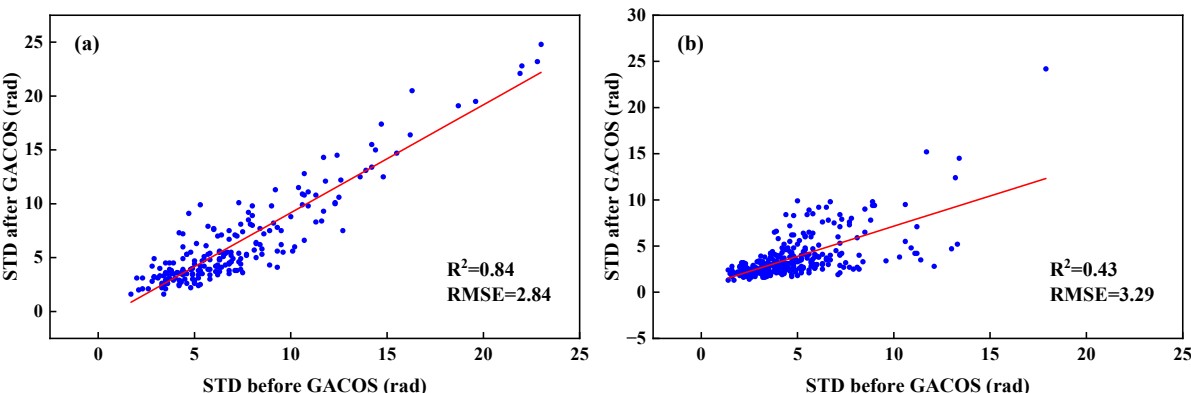

**Figure 5.** Comparison of phase standard deviation before and after atmospheric correction. ((**a**) for ascending orbit results, (**b**) for descending orbit results).

### 3.3. Phase Unwrapping Error Removal

The LiCSAR product processed by GAMMA SAR only employs the statistical cost flow SNAPHU method for phase unwrapping in the spatial dimension when unwrapping the interferometric phase, without considering the phase information in the time dimension [14,25]. Moreover, under the effect of the geometric distortion during radar imaging, some areas cannot conform to the phase continuity condition, such that the unwrapping time series and the accuracy of the deformation parameter solution will be severely affected. Accordingly, the unwrapped phase with large errors should be eliminated. In general, the LiCSBAS method comprises two steps in eliminating phase unwrapping errors. First, a comprehensive interferometric pair quality inspection is conducted, with the average coherence and effective pixel rate (the number of effective pixels in each interferogram/effective pixel total number) as the threshold to eliminate the interferometric pairs with low coherence and fewer effective pixels. Subsequently, a loop phase closure check is conducted for the interferograms that pass the quality inspection [24,25]. It is assumed that there are three

images $(\varphi_1, \varphi_2, \varphi_3)$ that can generate three pairs of unwrapped phases $(\varphi_{12}, \varphi_{13}, \varphi_{23})$, the loop closure phase is expressed in Equation (4):

$$\Delta_\varphi = \varphi_{12} + \varphi_{23} - \varphi_{13} \tag{4}$$

If there is no phase unwrapping error in the three interferograms, Equation (4) should approach 0. However, under certain factors (e.g., multi-view, filtering, and coherence variations), the loop closure phase is not entirely 0. If there is a phase unwrapping error in one or more interferograms, the loop closure phase should be an integer multiple of $2\pi$ [14,39]. LiCSBAS was adopted to calculate the loop closure phase and the root mean square error (RMSE) of the loop closure phase for the respective interferometric pair [24]. After identifying the loop closure phase with an RMSE larger than a predefined threshold (the threshold in this study was set to 1.5 rad) and conducting a small baseline (SBAS) inversion, it serves as the standard for masking pixels.

### 3.4. NSBAS Method to Invert Time Series Deformation Information in the Study Area

The NSBAS (the new small baseline subset) algorithm under the LiCSBAS technique was introduced to invert the time series deformation information of the study area to obtain the deformation time series of Baihetan Reservoir area from July 2019 to April 2022. Since the interferometric pairs formed have a small spatial vertical baseline, the residual terrain phase arising from the inaccurate external digital elevation model (DEM) is relatively small, such that the NSBAS algorithm was used directly to calculate the deformation rate and time series deformation of the respective pixel for the unwrapped phase [24,40,41]. It is assumed that there are N images, with acquisition times expressed as $(t_1, t_2, \cdots, t_N)$, and M interferometric pairs denoted as $(\varphi_1, \varphi_2, \cdots, \varphi_M)$, such that the interferometric phase of the respective pixel can be represented as shown in Equation (5):

$$\begin{cases} \varphi_{ij} = \sum\limits_{k-i}^{j-1} \Delta\varphi_k \\ \varphi_1 = 0 \end{cases} \tag{5}$$

where $\varphi_1$ denotes the interferometric phase of the first image in the N-image data set; $\varphi_{ij}$ represents the interferometric phase formed by images acquired at times $i$ and $j$; $\Delta\varphi_k$ expresses the phase increment between the $k-1$ and $k$th acquired images.

Notably, even if the fine network of interferograms is fully connected based on the respective pixel, time gaps may remain in the interferogram network. When there exist multiple subsets of interferograms, the coefficient matrix in Equation (5) may be rank deficient [40]. Although the singular value (SVD) decomposition method can still be employed for solving at this time, the minimum norm solution provided by SVD may have a bias, further affecting the calculated deformation rate while affecting the time characteristics of deformation information [41]. To obtain a more accurate and reliable displacement time series, since the deformation of the reservoir bank landslide can conform to the assumption of a long-term linear subsidence trend ($d = vt + c$) [42,43], the time constraint equation is introduced based on the NSBAS method based on Equation (5):

$$\begin{bmatrix} d \\ 0 \end{bmatrix} = \begin{bmatrix} \begin{bmatrix} G & 0 & 0 \end{bmatrix} \\ r\begin{bmatrix} 1 & 0 & \cdots & \cdots & 0 & -t_1 & -1 \\ \vdots & \ddots & \ddots & & \vdots & -t_2 & \vdots \\ 1 & \cdots & 1 & \ddots & \vdots & \vdots & \vdots \\ \vdots & & \vdots & \ddots & 0 & \vdots & \vdots \\ 1 & \cdots & 1 & \cdots & 1 & -t_{N-1} & -1 \end{bmatrix} \end{bmatrix} \begin{bmatrix} m \\ v \\ c \end{bmatrix} \tag{6}$$

where $d$ denotes the collection of unwrapped interferograms obtained at time $(t_1, t_2, \cdots, t_N)$; $r$ represents the weight factor of the time constraint equation; $G$ expresses the interfero-

metric pair connection matrix, composed of 0 and 1; $m$ is the Nth displacement increment vector matrix; $v$ is the deformation rate; $c$ is a constant.

The interferometric pairs after the GACOS atmospheric correction and phase unwrapping error removal in Section 3.3 were employed for small baseline inversion to obtain the surface time series deformation information of the Baihetan Reservoir area. The standard deviation of the deformation rate was estimated based on the Bootstrap method [44]. As indicated by the high estimated values of standard rate deviation, the surface deformation time series information contained noise or shows non-linear variations [24,44], and it can also characterize the measurement uncertainty of the deformation time series. Subsequently, the space-time filter (high-pass filter in time and low-pass filter in space) was adopted to separate the residual tropospheric noise, ionospheric noise, and orbit error noise pixels from the displacement time series, such that the time series of surface deformation and rate in the study area can be obtained (Figure 6).

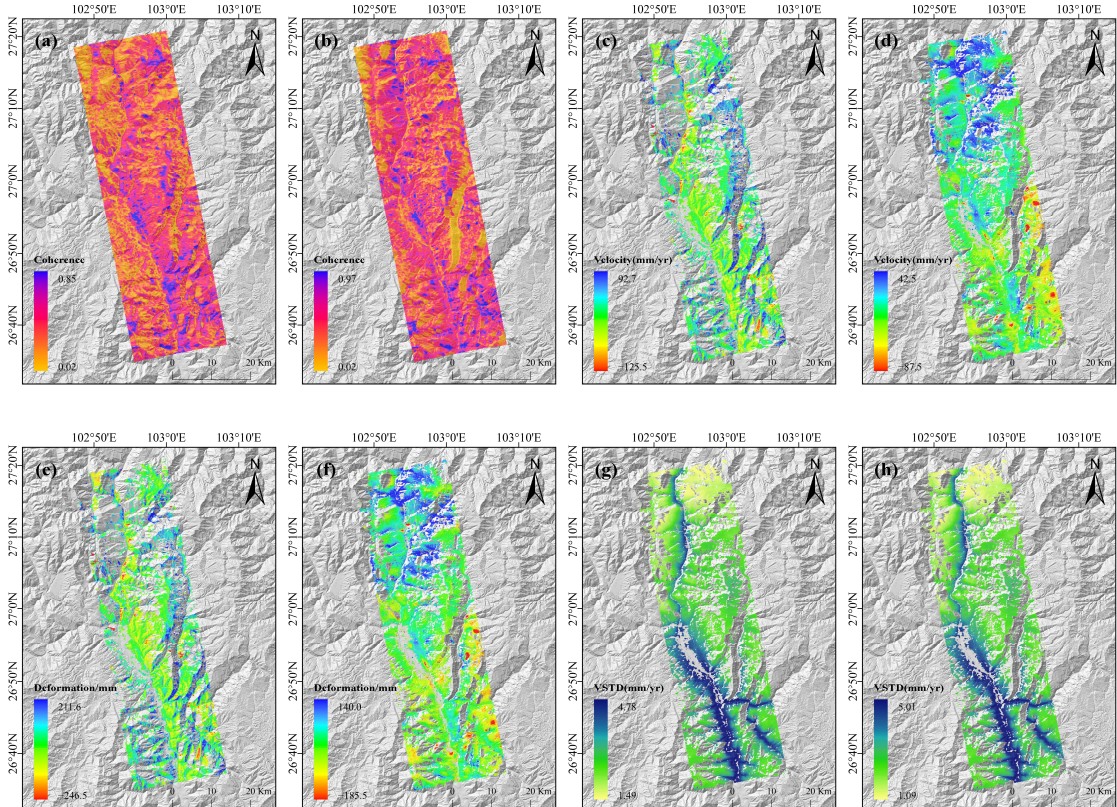

**Figure 6.** Time series information of surface deformation in the study area. ((**a**) represents the average coherence of ascending orbit data, (**b**) represents the average coherence of descending orbit data, (**c**) represents the annual average deformation rate of ascending orbit data, (**d**) represents the annual average deformation rate of descending orbit data, (**e**) represents the deformation quantity of ascending orbit data, (**f**) represents the deformation quantity of descending orbit data, (**g**) represents the standard rate deviation of ascending orbit data results, and (**h**) represents the standard rate deviation of descending orbit data results).

### 3.5. TLCC Model Analysis of Deformation Lag Effect

Hydrological factors are key inducers influencing reservoir bank landslide deformation, and the displacement rate of landslides has a certain lag to reservoir water level variations [5,45]. To quantitatively analyze the lag effect of the water level variations caused by the water storage stage and the deformation of the reservoir bank landslide in the Baihetan Reservoir area, the Time-Lag Cross-Correlation (TLCC) model [46] is introduced to analyze the water level variations and the deformation time series obtained by

the LiCSBAS technique. The TLCC model has been extensively used to characterize the numerical features of stochastic sequences and is an effective means to study the time lag relationship between two time series [46]. It has been widely employed in certain fields (e.g., acoustic ranging). For two discrete time series signals $x_1(k)$ and $x_2(k)$, their time-lag cross-correlation function is as follows:

$$R_{(n)} = \sum_{k=0}^{N} x_1(k) \cdot x_2(k+n) \tag{7}$$

where $k$ is the moment; $N$ denotes the total time; $n$ represents the delay time. If $x_1(k)$ and $x_2(k)$ are respectively the displacement time series of the reservoir bank landslide and the time series of reservoir water level variations caused by water storage, the lag response time $\tau$ of the displacement time series to the change of the reservoir water level can be obtained when the time-lag cross-correlation function $R_n$ obtains the delay time with the maximum value.

## 4. Results

### 4.1. Acquisition and Spatial Distribution Analysis of Ground Deformation Information

Figure 7 illustrates the time series ground deformation information (in the radar line-of-sight direction) of Baihetan Reservoir from July 2019 to April 2022 obtained using LiCSBAS technology. Here, positive values indicate moving towards the sensor direction, and negative values indicate moving away from the sensor direction. As revealed by the above result, the ground deformation information detected by the ascending and descending data sets did not completely correspond. The maximum deformation rate in the LOS direction of the ascending data was −125.5 mm/year whereas that of the descending data set reached −87.5 mm/year. The reason for this difference is affected by the side-looking imaging geometry of the Sentinel-1 satellite and the terrain fluctuation of the two banks of Baihetan Reservoir area. In general, the flight direction of the ascending data was roughly from southeast to northwest, and the radar line-of-sight direction was on the right, whereas that of the descending data was the opposite. Thus, for complex mountain areas where hydropower station reservoirs are located, deformation information detected by different orbit data should be complemented to avoid geometric distortion problems arising from single-orbit data, making the deformation detection results more comprehensive and accurate.

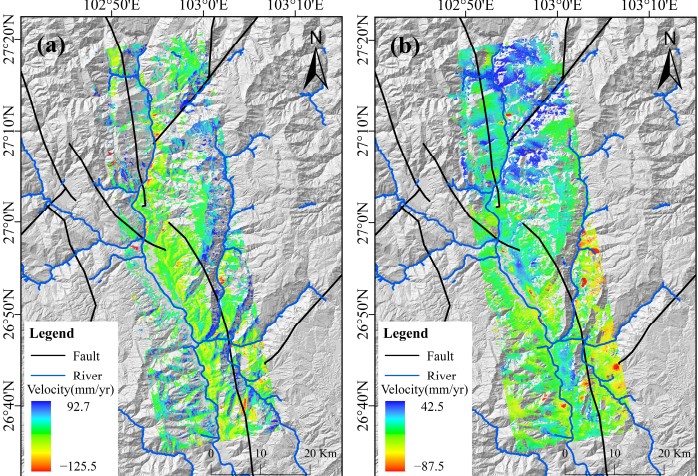

**Figure 7.** Ground deformation information in the radar line-of-sight direction in the study area. ((**a**) represents the deformation signals detected by ascending orbit data, (**b**) represents the deformation signals detected by descending orbit data).

Moreover, as depicted in Figure 7, the ground deformation results in the Baihetan Reservoir area show a clear spatial differentiation. Along the river flow direction of the Jinsha River Baihetan section, the ground subsidence areas were mostly concentrated in the upstream part of the reservoir, and the ground uplift areas were distributed in the downstream river section of the reservoir. The gradient change from high to low deformation was opposite to the direction of river flow. The time series ground deformation information of the research area was overlaid with faults and water systems for analysis (Figure 7) to explore the degree of correlation between this phenomenon and hydrogeological factors. As indicated by the analysis results, the development of ground deformation was highly correlated with faults and water systems. The upstream part of the reservoir area was primarily controlled by the Xiaojiang Fault Zone (Xundian section), with tributaries such as the Matree River and Yili River distributed. The downstream river section of the reservoir area develops the Xiaojiang Fault Zone (Dongchuan section), the Zhumu River Fault Zone, the Lianfeng Fault Zone, and the Ganluo-Zhuhe Fault Zone, and so forth. The areas with significant variations in ground deformation generally conformed to the distribution of the fault zone and water system, suggesting a certain correlation with hydrological and geological features.

### 4.2. Identification and Time Evolution Analysis of Reservoir Bank Landslide

The deformation signals detected by the combined ascending and descending data sets were used, and the characteristics of Baihetan Reservoir bank landslide were identified with the help of high-resolution optical images (Sentinel-2 and Google Earth). First, the deformation area was delineated mainly based on the deformation results obtained by LiCSBAS technology. To avoid the misjudgment of reservoir bank landslides attributed to the geometric distortion of single-orbit data sets, the deformation signals detected by different orbit data sets were superimposed and analyzed, and the areas with consistent deformation trends were retained as the interpreted deformation areas; then, optical images were employed to recognize reservoir bank landslides in conjunction with hue, structure, morphology, landslide boundary, and cracks, and other landslide characteristics. After a comprehensive recognition process combining InSAR deformation signals and optical image features, and also for further analysis of the lag effect of reservoir bank landslides, four typical reservoir bank landslides in the Baihetan Reservoir area were identified and selected (Figure 8), following the opposite direction of the Jinsha River flow, from north to south are the Miansha Village Landslide, Dawanzi Landslide, Wujiacun Village Landslide, and Wuli Landslide, sequentially numbered L1 to L4, respectively.

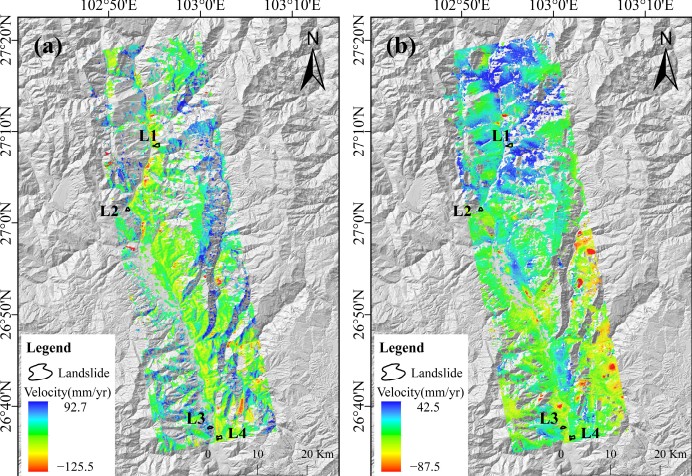

**Figure 8.** Reservoir bank landslide identification results. ((**a**) represents the identification results of ascending orbit data, (**b**) represents the identification results of descending orbit data).

The field surveys of typical reservoir bank landslides that were identified using LiCS-BAS technology were conducted through Drone photogrammetry. Figure 9 presents the results of the field surveys of four typical reservoir bank landslides (the L4 landslide body was not subjected to drone aerial imaging but only field observation, and it was replaced by Google Earth high-resolution optical image), thus suggesting significant landslide features (e.g., boundaries, flanks, cracks, strong deformation areas, and steep banks). The L1 landslide, located in Miansha Village in the downstream reservoir area on the east bank of the Jinsha River, exhibits an irregular tongue-shaped morphology, significant depression features on the slope, as well as broken surface rock bodies. As indicated by the result of the field investigation, debris was moving on the surface, suggesting that the landslide body is very likely in a state of long-term sliding. The L2 landslide was located in the middle of the reservoir area on the west bank of the Jinsha River at the Dawanzi tunnel. The slope body was irregularly pear-shaped, the upper part was covered with sparse vegetation, and sliding traces existed at the lower edge of the slope body in an exposed state. The field investigation suggested that this landslide developed ground cracks, the deformation displayed an uneven spatial distribution, and fan-shaped deposits were formed at the bottom of the landslide. The L3 landslide was located in the upstream western bank of the reservoir area in Wujiacun Village. To be specific, it was irregularly tongue-shaped, with significant depression features of the slope body. As revealed by the field investigation, the deformation of this slope body was primarily attributed to squeezing deformation from the edge to the center, and cracks and debris existed in the middle of the landslide. The L4 landslide was located on the downstream eastern bank of the reservoir area at Wuli Slope. The slope body was irregularly shaped like a dustpan. The field investigation indicated that fan-shaped deposits were formed at the bottom of the slope body, the deformation displayed an uneven spatial distribution, and the strong deformation area was located in the middle and lower parts of the slope body.

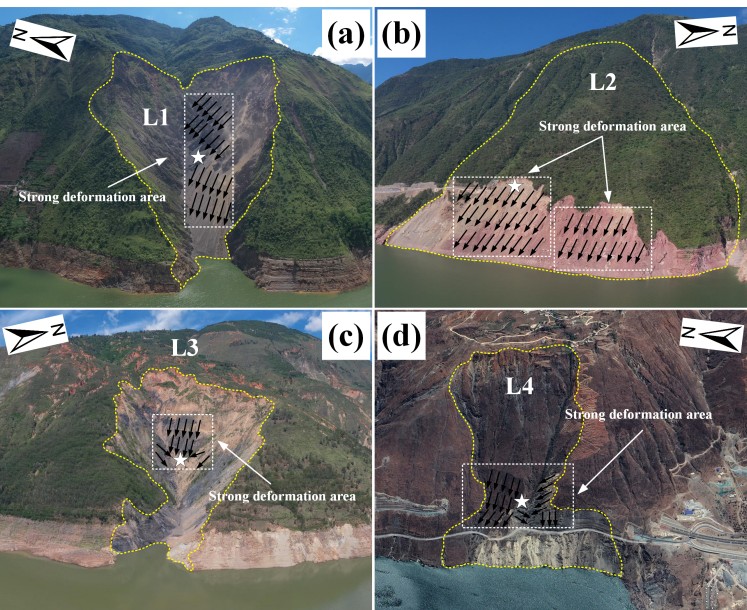

**Figure 9.** Field survey results of typical reservoir bank landslides. ((**a**) represents L1 landslide, (**b**) represents L2 landslide, (**c**) represents L3 landslide, (**d**) represents L4 landslide, the white pentagrams represent the feature points selected in the high deformation area, while the black arrows indicate the local sliding direction of the landslide.).

Deformation characteristic points were selected in the strong deformation area of typical reservoir bank landslides (Figure 9 for the position of the characteristic points) to analyze the time evolution law of reservoir bank landslide deformation. The deformation time series curve was drawn (Figure 10). As depicted in the figure, the typical reservoir bank

landslides selected from the upstream, middle, and downstream of Baihetan Reservoir area were all in a state of long-term sliding. The deformation time curves contained significant seasonal fluctuations, and the cumulative displacement exceeded 15 mm in less than two and a half years. As indicated by the above result, the creep of the reservoir bank landslide largely arose from hydrological events and weathering [46,47]. The alternating dry and wet climate, steep terrain, and water storage in the Baihetan Reservoir area contributed to the development of landslide creep. Moreover, the comparison of before and after the water storage stage of Baihetan Reservoir area (blue gradient area in Figure 10) suggested that the deformation trend of the reservoir bank landslide changed after the water storage. Although it still maintained an overall fluctuation trend, the deformation gradient variations to a certain extent, which can be likely to be correlated with the variations in the water level after the reservoir storage.

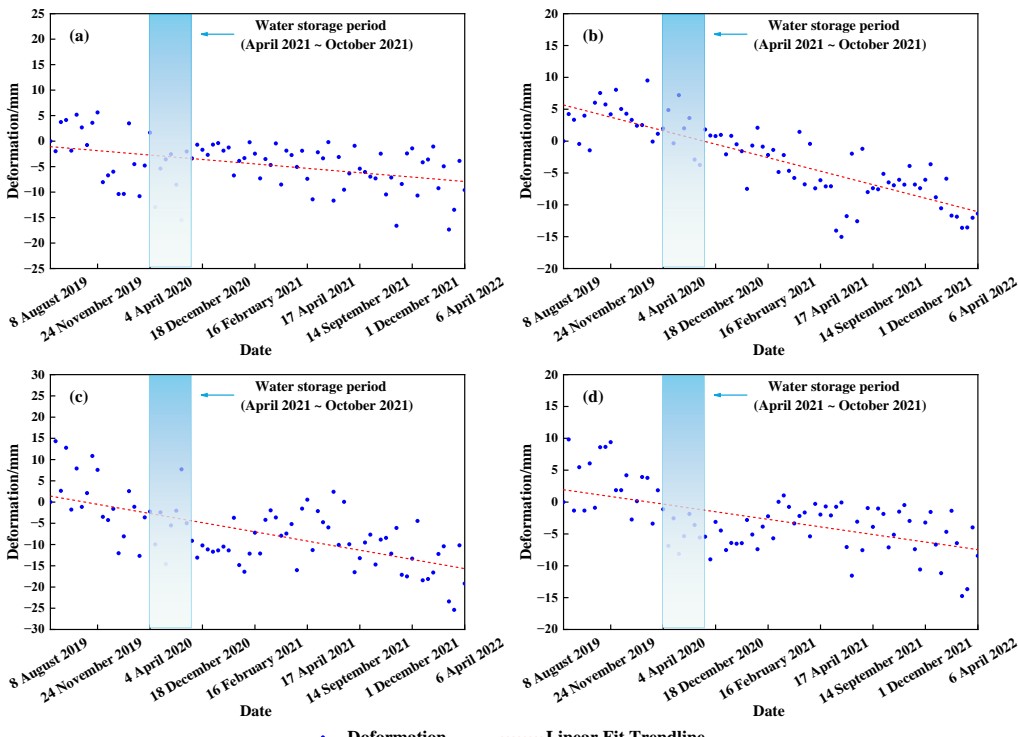

**Figure 10.** Time series deformation curve of the deformation feature points of typical reservoir bank landslides. ((**a**) represents L1 landslide, (**b**) represents L2 landslide, (**c**) represents L3 landslide, (**d**) represents L4 landslide. The blue gradient square area in the figure represents the large-scale water storage in the Baihetan Reservoir from April to October 2021. The red dashed line represents the linear fitting of the time series deformation of the reservoir bank landslide).

### 4.3. Quantitative Analysis of the Lag Time of Reservoir Bank Landslide Deformation

Relevant research suggested that hydrological factors are important factors for the deformation of reservoir bank landslides [45–47]. To gain more insights into the lag effect of reservoir bank landslide deformation in the Baihetan Reservoir area, the deformation time series during the water storage stage (April 2021 to October 2021) was cut from Figure 10, and the time series curve of water level variations and rainfall during the reservoir water storage was generated (Figure 11). As depicted in Figure 11, the increase in the water level at the phase of water storage and rainfall changed synchronously, the rainfall effectively replenished the storage capacity of Baihetan Reservoir, such that favorable conditions were created for the increase in the water level. However, in the period of reservoir water storage, the deformation of reservoir bank landslides displayed significant fluctuating variations. The cumulative displacements of the four selected typical landslides exceeded 10 mm, and the deformation curve was not consistent with the increase in the reservoir water level,

suggesting that the reservoir bank landslides in the Baihetan Reservoir area show a notable lag effect with the increase in the water level at the phase of water storage. The lower edge of the landslide was subject to the seepage and erosion arising from the increase in the water level, and other reservoir bank reconstruction actions had a certain lag time. To quantitatively analyze the above lag time, the time series of typical landslide characteristic point deformation and reservoir water level variations from April 2021 to October 2021 were calculated using the TLCC model. Figure 12 presents the calculation results. As depicted in Figure 12, the lag times of the Miansha Village landslide, the Dawanzi landslide, Wujiacun Village landslide, and Wuli landslide responding to the water level increase in the Baihetan Reservoir reached 5 d, 7 d, 6 d, and 7 d, respectively, with an average lag time of nearly 6 d, and the correlation coefficients exceeded 0.45 (the correlation coefficients of the characteristic points of other landslides exceeded 0.5, with the exception of the characteristic points of Wu Family Village landslide), and they passed the 95% significance test, showing a significant correlation. As revealed by the above-mentioned result, the lag time for reservoir water to seep into and erode the reservoir bank landslides, leading to the sliding of reservoir bank landslides due to variations in reservoir water level, reached 5–7 d after the Baihetan Reservoir storage.

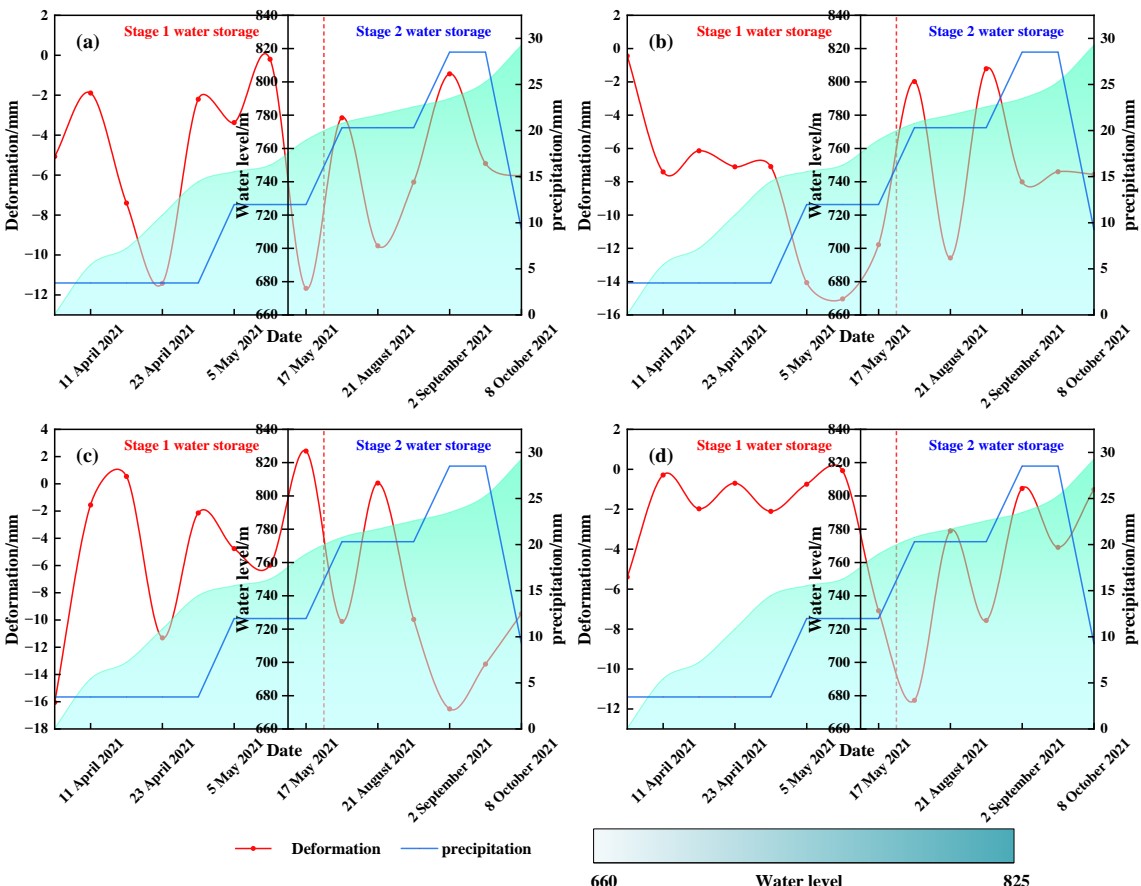

**Figure 11.** Time series curves of typical landslide feature points, rainfall, and water level increase during the water storage stage. ((**a**) represents L1 landslide, (**b**) represents L2 landslide, (**c**) represents L3 landslide, (**d**) represents L4 landslide. The red curve represents the temporal deformation sequence of the reservoir bank landslide during the reservoir impoundment period. The blue curve represents the change in monthly average rainfall during the reservoir impoundment period. The light green gradient area represents the change in reservoir water level during the reservoir impoundment period. The red dashed line represents the time interval between the two stage impoundments within the complete reservoir impoundment period at Baihetan reservoir area).

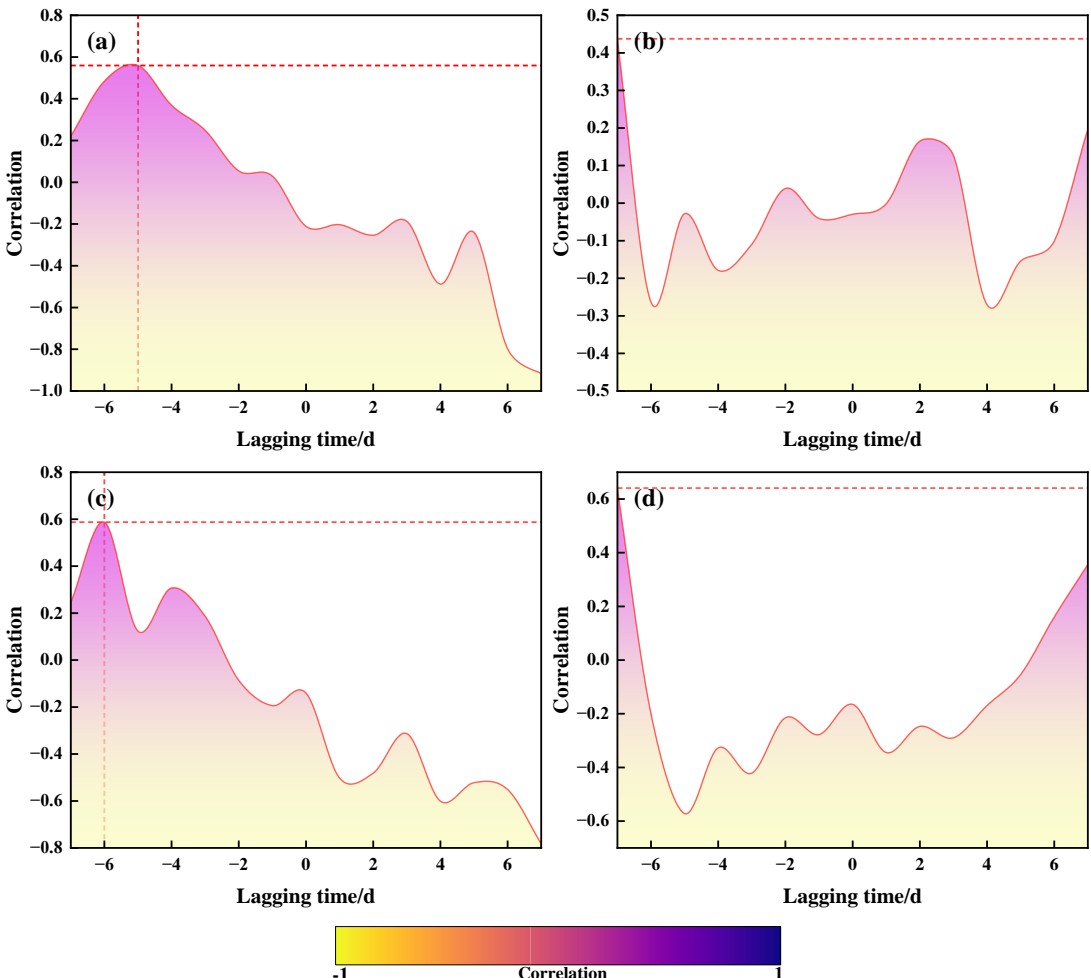

**Figure 12.** Time lag cross-correlation sequence of deformation-reservoir water level increase. ((**a**) represents L1 landslide, (**b**) represents L2 landslide, (**c**) represents L3 landslide, (**d**) represents L4 landslide).

## 5. Discussion

### 5.1. Evaluation of LiCSBAS Technology Measurement Accuracy

In this study, LiCSBAS technology was adopted to obtain the time series deformation information of the Baihetan Reservoir area, which was limited by the LiCSAR product released by the COMET project of the NERC Earthquake and Volcano Structure Observation and Modeling Center in the UK. Compared with ALOS PALSAR, TerraSAR-X-1, or other data, LiCSBAS technology can more significantly apply to coping with Sentinel data [24]. In this study, the surface deformation time series of the study area was mainly obtained based on LiCSAR data and input into the TLCC model to quantitatively analyze the lag time of reservoir bank landslides. Accordingly, the accuracy of InSAR technology monitoring deformation results directly affects the accuracy of the quantitative analysis of lag effects. Due to the lack of level measurement and GPS measurement data in the Baihetan Reservoir area during the research period, we compared the InSAR results of this study with some published InSAR research results during the approximate research period and summarized the main differences in Table 3. It can be seen that although the InSAR results of this study have some similarities with the InSAR results in Table 3, there are certain differences in quantitative values. The main reason for this difference is that the research period of Dun [13] and Wu [33] was mainly before the reservoir storage, and does not include deformation information after storage. The research results of Dai et al. [48] merged the deformation signals of ascending and descending tracks, and this study's InSAR time series,

in addition to including the period before storage and during reservoir storage, also used ring phase closure difference checks and GACOS atmospheric data to weaken some errors affecting the accuracy of InSAR results. In addition, to further evaluate the accuracy of the deformation signal detected by LiCSBAS technology in this study, we computed and analyzed the distribution trends of standard deviations of velocity points from ascending and descending data sets within the study area (Figure 13). As depicted in Figure 13, the velocity standard deviations from different orbital data are all less than 5 mm/year and conform to a normal distribution. The probability density of velocity standard deviations exceeding 3 mm/year is concentrated below 0.05, validating the accuracy of the InSAR results in this study. It also underscores that reducing atmospheric delay errors and phase unwrapping errors during InSAR surface deformation monitoring can effectively improve the precision of the final results.

**Table 3.** Main differences in comparative research.

| Comparative Study | Dun, et al., 2022 [13]. | Wu, et al., 2022 [33]. | Dai, et al., 2023 [48]. | This Study |
|---|---|---|---|---|
| SAR data | Sentinel/PALSAR | Sentinel | Sentinel | LiCSAR (Sentinel) |
| Orbital | Ascending | Ascending | Ascending | Ascending |
| direction | Descending | Descending | Descending | Descending |
| Processing Software | SARscape | SARscape | SARscape | LiCSBAS |
| Time span | October 2014~August 2020 | April 2014~March 2021 | April 2020~October 2021 | July 2019~April 2022 |
| Ascending orbit (mm/year) | −78~67 | −41~68 | −132~50 (Before water storage) | −126~93 |
| Descending orbit (mm/year) | −56~10 | −24~46 | −145~45 (After water storage) | −88~43 |

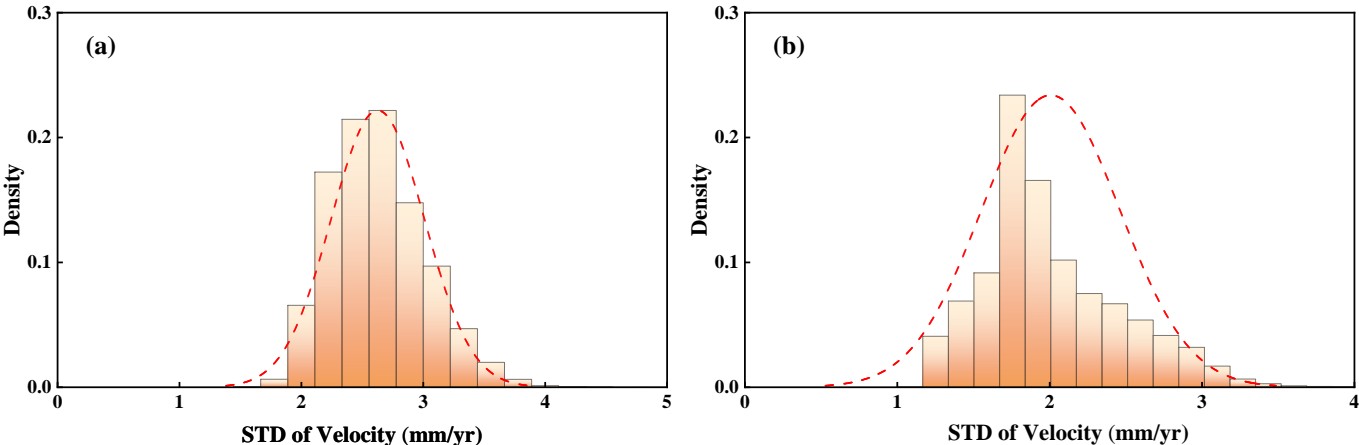

**Figure 13.** Standard deviation of Velocity (VSTD), (**a**) represents the ascending orbit results, (**b**) represents the descending orbit results, the red dashed line represents the fitting curve of the normal distribution, and the color transition from dark to light in the bar graph signifies a change in density from low to high.

## 5.2. Analysis of Reservoir Bank Landslide Lag Effect

Certain studies suggest that deformation of reservoir bank landslides, triggered by hydrological events (primarily including rainfall and water level fluctuations), is primarily due to variation in pore water pressure on the landslide slip surface. This variation is caused by changes in water levels [45–47]. However, during the storage phase, stability of the reservoir bank landslide is almost entirely regulated by water level fluctuations [45,46,49–51], rendering the impact of rainfall negligible. Additionally, under varying storage conditions, deformation signals of reservoir bank landslides exhibit different time-lag response patterns. Interestingly, the response of the deformation of reservoir bank landslides in the Baihetan reservoir area to the elevation in the reservoir water level during the storage phase is not instantaneous. There exists a time lag between the deformation peak and the reservoir water level peak, a phenomenon consistent with previous studies, suggesting a lag effect of reservoir bank landslides to water level variations within the reservoir area. The lag time for the response to the rise in the reservoir water level for the Miansha Village

landslide, Dawanzi landslide, Wujiacun landslide, and Wuli landslide, determined from a quantitative analysis using the TLCC model, are 5 d, 7 d, 6 d, and 7 d, respectively. The average is around 6 d, showing a distinct disparity compared to previous studies on landslide deformation response to periodic water levels. Huang and other researchers [28] studied the Xinpuzi landslide in the Three Gorges reservoir area. They found the landslide's response lag time to the reservoir water level to be approximately 31 d. Wen et al. [45] discovered that the deformation lag time of the reservoir bank landslide at the Maoergai Hydropower Station was around 65~120 d. This quantitative numerical difference is very likely connected with the different time lag response patterns of the deformation signals of the reservoir bank landslides under varying storage conditions. Since April 2021, the Baihetan reservoir area has been impounding water. By October 2021, the reservoir water level had rapidly increased from 660 m to 812 m [33,48]. The large-scale and high-intensity storage mode is a significant factor inducing the rapid response of the reservoir bank landslides in this study. The difference in research periods also plays a crucial role in the observed discrepancy. Huang and Wen et al. focused their study on the response time of deformation to periodic water level variations. Conversely, this study aims to uncover the lag phenomenon of reservoir bank landslides deformation during the storage period in response to the rise in water level. Contrary to the impact of periodic water level changes on reservoir bank landslides, during the storage period, the originally exposed lower edge of the slope is subjected to river erosion [45]. The soil structure of the lower slope tends towards a loose state under the action of dynamic water pressure. The balance between the upper and lower edges of the slope changes, and the groundwater level inside the reservoir bank landslide gradually rises [9]. This leads not only to a tendency towards saturation within the landslide, but also increases the hydrostatic pressure on the slope surface, reducing the effective stress of the soil and the shear strength of the slip surface. This forces the landslide into an unstable state and naturally triggers rapid deformation [45,52]. Therefore, the reservoir bank landslides in the Baihetan reservoir area exhibit a shorter time-lag response during the storage period compared to the period of periodic water level changes. This understanding provides a vital supplement to the more accurate study of the response pattern of reservoir bank landslide deformation to water level changes.

*5.3. Analysis of the Causes of Reservoir Bank Landslide Deformation Lag*

When the water level in the reservoir varies, the water content within the reservoir bank landslide requires a certain time to reach a new equilibrium. The reason for this result is that the change in pore water pressure in the landslide can exert a certain effect on the stability of the soil, and the stress state of the soil should also adjust and adapt to new pressure conditions, both of which can result in a lag effect in the reservoir bank landslide. Relevant research has suggested that the lag effect comprises the time delay in the process of landslide deformation and the irreversible plastic deformation of the landslide geotechnical structure during the process of water level variations. Reservoir bank landslides exhibit complex origins and varied types. Since the deformation process of reservoir bank landslides can be based on the hypothesis consisting of seasonal non-linear deformation and long-term linear subsidence trends, their lag effect on reservoir water level variations may be correlated with the external environment of the landslide (e.g., elevation, slope, aspect, and vegetation coverage) and the internal structure of the soil (e.g., lithology). To explore whether the lag effect of the reservoir bank landslides in the Baihetan reservoir area on the water level variations during the storage period arises from the above reasons, the factors were extracted (e.g., elevation, slope, aspect, lithology, and vegetation coverage of the Miansha Village landslide, Dawanzi landslide, Wujiacun landslide, and Wuli landslide) (Table 4). As depicted in this table, the vegetation coverage of the selected typical landslides was all below 0.2, i.e., low vegetation coverage, probably correlated with the slow creep of the landslide. As indicated by the comparison of the geological attributes of typical landslides, although the Dawanzi landslide and Wuli landslide had the same lag time to the water level, their aspect and lithology were

notably different. Although the Miansha Village landslide, the Dawanzi landslide, and the Wujiacun landslide all exhibited the lithology of mixed sedimentary rocks with small slope differences, the lag time was positively correlated with vegetation coverage and negatively correlated with aspect. Furthermore, it is noteworthy that the Miansha Village landslide and Wuli landslide are located on the east bank of the Jinsha River, whereas the Dawanzi landslide and Wujiacun landslide are on the west bank of the Jinsha River. However, their lag response patterns to water level variations were different, probably correlated with the Coriolis force due to Earth's rotation. The Baihetan section of the Jinsha River flowed approximately from south to north, and the river water, affected by the rotational velocity of the Earth, tended to erode more on the east bank of the Jinsha River. Thus, the Miansha Village landslide located on the east bank exhibited the shortest lag time among all typical landslides.

**Table 4.** Geological attributes of typical reservoir bank landslides.

| Landslide (Number) | Lagging Time/d | Elevation/m | Slope/(°) | Aspect/(°) | FVC | Rockiness |
| --- | --- | --- | --- | --- | --- | --- |
| Miansha Village (L1) | 5 | 1231 | 48.43 | 200.53 | 0 | Mixed sedimentary rocks |
| Dawanzi (L2) | 7 | 869 | 37.19 | 106.43 | 0.19 | Mixed sedimentary rocks |
| Wujia Village (L3) | 6 | 808 | 41.93 | 160.63 | 0.10 | Mixed sedimentary rocks |
| Wuli (L4) | 7 | 964 | 33.24 | 288.69 | 0.14 | Acidic deep-formed rocks |

## 6. Conclusions

In this study, a method of combining LiCSBAS technology and TLCC model was proposed to monitor the deformation of reservoir bank landslides and quantitatively analyze their lag characteristics, such that some issues currently present in the monitoring and lag analysis of reservoir bank landslide deformation can be addressed. First, the LSM algorithm and R index were adopted to identify geometric distortion phenomena (e.g., shadow, layover, and foreshortening in the Baihetan reservoir area) to analyze geometric distortion areas of SAR data and mask the shadow area. Subsequently, the deformation time series of reservoir bank landslides were built, and the spatial distribution characteristics and temporal evolution rules of reservoir bank landslide deformation in Baihetan Reservoir were analyzed based on 576 ascending and descending orbit LiCSAR data from July 2019 to April 2022 and LiCSBAS technology. Lastly, the TLCC model was used to quantitatively analyze the lag effect of the deformation time series of reservoir bank landslides on the variations in reservoir water level at the reservoir storage phase in the Baihetan reservoir area. The conclusions of this study are drawn as follows:

(1) LiCSBAS technology is capable of effectively monitoring the mountainous surface deformation in the Baihetan reservoir area. The maximum surface deformation rate obtained by LiCSBAS technology in the Baihetan reservoir area from July 2019 to April 2022 was $-125.5$ mm/year. The standard deviations of velocity from various orbital data were all less than 5 mm/year, and the probability density for the standard deviation of velocity exceeding 3 mm/year was concentrated below 0.05, consistent with the error distribution rule. The surface deformation results of the Baihetan reservoir area exhibited significant non-linear spatial distribution characteristics. Following the river direction of the Baihetan section of the Jinsha River, the surface subsidence areas were primarily concentrated in the upstream part of the reservoir area, and the surface uplift areas were distributed in the downstream river sections. The gradient change of surface deformation from high to low was opposite to the river direction, and the distribution was correlated with the distribution of faults and water systems in the study area to a certain extent.

(2) The reservoir bank landslides in the Baihetan reservoir area detected by LiCSBAS technology and assisted by optical image identification were at a long-term sliding state. Significant landslide features (e.g., boundaries, flanks, cracks, strong deformation zones, and steep slopes) can be identified from field surveys. The time-series deformation curve covered significant seasonal fluctuations, and the selected typical reservoir bank landslides showed accumulated displacements exceeding 15 mm in less than two and a half years.

Moreover, the deformation trend of reservoir bank landslides before and after reservoir impoundment varied notably, suggesting that variations in reservoir water level during the impoundment stage can serve as the major factor inducing landslide instability.

(3) The deformation of reservoir bank landslides in the Baihetan reservoir area did not vary synchronously with the increase in the reservoir water level at the impoundment stage, whereas it exerted a significant lag effect. The lag response time of the deformation time series of reservoir bank landslides and the time series of reservoir water level variations was nearly 5–7 days, with an average lag time of approximately 6 days. The landslide with the shortest lag response time was located on the east bank of the reservoir, i.e., an important addition to the research on the response mode of reservoir bank landslide sliding to water level variations. The lag response mode of the reservoir bank landslides in the Baihetan reservoir area may be significantly correlated with the landslide's elevation, slope, aspect, lithology, vegetation coverage, and the Coriolis force arising from the rotation of the Earth. Furthermore, the lag time of some typical reservoir bank landslides was positively correlated with vegetation coverage and negatively correlated with aspect to a certain extent.

**Author Contributions:** Formal analysis, Z.Y. (Zhengrong Yang) and W.X.; Funding acquisition, W.X. and Z.Y. (Zhiquan Yang); Methodology, Z.Y. (Zhengrong Yang) and G.H.; Project administration, W.X.; Resources, Z.S.; Software, Z.Y. (Zhengrong Yang), J.G. and D.Y.; Supervision, Z.Y. (Zhiquan Yang); Writing—original draft, Z.Y. (Zhengrong Yang); Writing—review and editing, Z.Y. (Zhengrong Yang), W.X., Z.Y. (Zhiquan Yang), Z.S., G.H., J.G. and D.Y. All authors have read and agreed to the published version of the manuscript.

**Funding:** This research was funded by the National Natural Science Foundation of China (Grant No. 41861134008), Muhammad Asif Khan academician workstation of Yunnan Province (Grant No. 202105AF150076), General program of Yunnan Province Science and Technology Department (Grant No. 202105AF150076), Key Project of Natural Science Foundation of Yunnan Province (Grant No. 202101AS070019), Key R&D Program of Yunnan Province (Grant No. 202003AC100002), General Program of basic research plan of Yunnan Province (Grant No. 202001AT070059), Major scientific and technological projects of Yunnan Province, Research on Key Technologies of ecological environment monitoring and intelligent management of natural resources in Yunnan (Grant No. 202202AD080010), "Study on High-Level Hidden Landslide Identification Based on Multi-Source Data"of Key Laboratory of Early Rapid Identification, Prevention and Control of Geological Diseases in Traffic Corridor of High Intensity Earthquake Mountainous Area of Yunnan Province (Grant No. KLGDTC-2021-02), Guizhou Scientific and Technology Fund (Grant No. QKHJ-ZK [2023] YB 193), and Yunnan Normal University Graduate Student Research Innovation Fund (Grant No. YJSJJ23-B151).

**Data Availability Statement:** Not applicable.

**Acknowledgments:** The authors would like to thank COMET for providing the LiCSAR data free of charge, and JXAX for providing the 30 m DEM free of charge.

**Conflicts of Interest:** The authors declare no conflict of interest.

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
