# Peer review of "Time-Lag Response of Landslide to Reservoir Water Level Fluctuations during the Storage Period: A Case Study of Baihetan Reservoir"

_water, doi:10.3390/w15152732_

Round 1

Reviewer 1 Report

A method of combining LiCSBAS technology and TLCC model was proposed to monitor the deformation of reservoir bank landslides and quantitatively analyze their lag characteristics, such that some issues currently present in the monitoring and lag analysis of reservoir bank landslide deformation can be addressed. This study is interesting. I recommended accepting it after minor revision. The review comments are as follows:

1. Please discuss the accuracy of the results

2. Please state the innovation of this paper in the literature review

3. The font of the table is too large, please modify it

Reviewer 2 Report

Dear Authors,

I read your exciting manuscript and find it a very suitable and important contribution to Water (MDPI). The research design is fine, but many explanations are long and convoluted, hence I suggest some simplification in expressions used. I find very novel how you used variable datasets and combined their results into a coherent model. While this is great, again, some convoluted and complex expressions preventing to let the manuscript more direct or better explained its global relevance. I also find some technical issues like far better utilization of figure captions, especially in important but complex figures in the last pages of the manuscript. I think the figures are great, but they have to have well developed captions allowing the reader to understand the contents from the captions and figure alone.

I have provided an annotated PDF where I marked my suggestions.

Best regards,

English is fine, some minor editing needed.

Reviewer 3 Report

The Manuscript entitled "Time-lag Response of Landslide to Reservoir Water Level Fluctuations during the Storage Period: A Case Study of Baihetan Reservoir" is interesting scientific work, that can be published after some revision, following the recommendations:

Section 2.1.1. It would be beneficial to present the map of the place in figure

- Please carrefully check the template editing (e.g. page no. 10)

- Please add correlation parameters into figure 5. 

- Please add information into the figure, how long was the water storage period. 

- please explain in some cases sets of references e.g. line 564 [3,23,40-42] whats the differenece and why all of them are reffered to?
